# Addressing Cognitive Bias in Adolescents with Neurodevelopmental Disorders Using 3-D Animated Serious Games

**DOI:** 10.3390/pediatric17020028

**Published:** 2025-02-25

**Authors:** Suzanne Stewart, Stephen John Houghton, Leslie Macqueen

**Affiliations:** The Graduate School of Education, The University of Western Australia, Perth 6009, Australia; suzanne.stewart@uwa.edu.au (S.S.); leslie.macqueen@uwa.edu.au (L.M.)

**Keywords:** serious games, negative interpretive bias, neurodevelopmental disorders, adolescents, mental health, intervention

## Abstract

Objective: This study sought to evaluate the effectiveness of a serious game, that embeds cognitive bias modification for interpretation (CBM-I), in altering the negative interpretive bias of early aged adolescents diagnosed with Attention-Deficit/Hyperactivity Disorder, Autism Spectrum Disorder, and Specific Learning Disorders. The difficulties that adolescents with neurodevelopmental disorders (NDDs) experience navigating the social nuances of everyday environments make them prone to the cognitive biases that lead to the development of negative thought patterns. Directly tackling the biased interpretive processes that give rise to negative thinking may be effective in reducing negative bias and mental health problems. Method: Minds Online, a 10-episode 3-D animated serious game that embeds CBM-I was introduced using a three-phase multiple baseline design in a school setting. Eight adolescents diagnosed with an NDD completed the 10 episodes. Results: Real-time data revealed that seven of the eight adolescents altered their negative interpretive bias to a benign bias. However, pre- and post-test standardized measures revealed non-significant changes in the desired direction for mental health. Visual analyses of 308 daily self-reported ratings about worry about schoolwork, worry about peer relationships, and feelings of loneliness did not demonstrate a replicated intervention effect. However, when these interrupted time series data were analyzed statistically, significant individual improvements were evident. Engagement with Minds Online was excellent, as was adherence to daily data collection. Conclusions: Minds Online seems to be highly effective in altering the negative interpretive biases of adolescents with NDDs, which is promising because such cognitive biases are involved in the onset and maintenance of psychopathology.

## 1. Introduction

Neurodevelopmental disorders (NDDs) affect 3–7% of children and adolescents globally [1,2,3], although some estimates are as high as 15–20% [4,5]. Known for their chronic and heterogenous nature [6,7], Attention-Deficit/Hyperactivity Disorder (ADHD), Autism Spectrum Disorder (ASD), and Specific Learning Disorders (SLD) are the most commonly presenting NDDs in child and adolescent mental health services (CAMHS) and mainstream schools worldwide [8].

The high comorbidity within and between NDDs, especially ADHD, ASD, and SLD, can affect the magnitude of functional outcomes for children and adolescents [9] and can heighten the risk of developing mental health problems [10]. For example, rates of depression and depressive symptoms are much greater in those with ADHD [11,12] and highly prevalent in young people with ASD [13,14,15] and SLD [16]. It is unsurprising therefore that NDDs represent one of the leading causes of lifelong disabilities [17] and are increasingly emerging as the primary cause of morbidity in children [18,19].

During adolescence, peer relationships intensify and become more complex [20]. For adolescents with NDDs, the deficits they experience in social cognition [21] affect their ability to understand and respond to social interactions, a critical skill for adjusting their own behavior and fostering reciprocal relationships [22]. It also leads to misinterpretations of others’ emotions, intentions, and beliefs [23]. As a result, increased difficulties with social interactions that limit the quantity or quality of friendships they develop is experienced by adolescents with ADHD [24], ASD [14,25], and comorbid SLD [26]. This leads to an increased risk of mental health problems in adolescents with ADHD [12], ASD [14,15,26], and SLD [16]. For many adolescents with NDDs, feelings of disconnection, social isolation, and loneliness are subsequently felt [27].

These challenges, alongside a tendency to worry, cause an increased risk of developing negative thought patterns and cognitive distortions, particularly when interpreting everyday ambiguous interpersonal stimuli [28,29]. There is now substantial evidence that adolescents who report high levels of anxiety (such as those with NDDs) tend to impose negative resolutions on ambiguous information [28,30]. This interpretive bias contributes to worry [31], which is a key cognitive indicator and causal factor in anxiety disorders and other mental health disorders [32,33,34,35]. Adolescents with ADHD, ASD, and SLD may be more prone to cognitive biases (that precede worry) and, as such, more vulnerable to mental health problems [36], social withdrawal, and loneliness [37].

The increasing rates of mental health problems and loneliness amongst children and adolescents has been identified as a critical global health issue in the aftermath of the coronavirus disease 2019 crisis [38] and the public mental health system has struggled to meet the growing demand [39]. Schools have been highlighted as ideal places for cost effective interventions that can be delivered en masse [40] and which can alter young people’s maladaptive interpretations and lead to lower frequencies of mental health problems [41,42]. In favor of this is that schools have various technologies, and evidence shows that increasing mental health treatment engagement via technology delivered interventions is associated with improvements in mental health see [43].

## 2. Cognitive Bias Modification for Interpretation (CBM-I) and Serious Games

One promising approach that uses technology (e.g., smartphone app, computer), addresses ambiguity, can be delivered by non-clinicians in familiar environments such as classrooms, and reduces the stigma around seeking help by self-conscious young people is CBM-I [44]. CBM-I builds upon cognitive theory in that how we think and explain events affects our emotional and behavioral responses [45]. By altering biased interpretations at an early stage, CBM-I prevents negative, distorted thinking before it develops into the conscious thoughts that lead to dysfunction [28]. Although evidence shows this approach can effectively reduce negative thought intrusions (from a single session to six weeks), e.g., ref. [46], contemporary CBM-I formats are ‘inherently boring’ ([47], p. 14), particularly for young people.

The use of serious games (SG) for the delivery of short- and long-term psychological interventions offers immense potential [48] for innovative, novel, engaging [49], and evidence-based techniques [48] to be delivered as therapies for young people. A growing number of studies have focused on the use of SGs to assist people with NDDs, especially those with ASD and ADHD [50,51,52,53]. However, adolescents are more likely to be receptive to interventions that address daily experiences [54], and current CBM-I approaches do not enable this.

To date, there appears to be no serious game-based interventions that incorporate daily experiences to directly tackle the biased interpretive processes that give rise to negative distorted thinking, especially among more vulnerable populations such as adolescents with NDDs. Because adolescents with NDDs are socially reticent in face-to-face situations and may not have the level of introspection and ability to be conscious of their own negative thoughts (which they must be able to evaluate and resolve), an engaging serious game intervention may be a viable approach to help these adolescents learn more adaptive interpretative styles that enhance mental health and wellbeing. Minds Online is a serious game intervention and advances the field by offering such an innovative, novel, and engaging approach for delivering a therapy to adolescents with NDDs. Furthermore, by incorporating daily real-life events in game play, Minds Online provides participants with multiple opportunities to visualize themselves experiencing the scenario in a non-negative interpretational tendency. That is, it provides a more adaptable and functional approach to adolescents in the environment in which they operate and experience life events (e.g., school) and where cognitive and behavioral responses must be intuitive see [45].

This current study introduces Minds Online, a 3-D animated serious game that embeds CBM-I, to help adolescents resolve negative thought patterns that arise from their daily academic and social experiences. Described in detail in Section 3, Minds Online was co-designed over three years with adolescents aged 10 to 16 years, their parents, teachers, and school psychologists. The main objective of this study is to evaluate the effectiveness of Minds Online with a sample of early aged adolescents diagnosed with ADHD, ASD, and SLD.

## 3. Materials and Methods

### 3.1. Participants and Setting

The sample consisted of nine, Year 6, 11- to 12-year-old mainstream school adolescents formally diagnosed with an NDD. Of the sample, eight participated in the intervention program. One participant, who initially agreed to be involved but subsequently decided not to take part, became a Constant Series Control (CSC), thereby providing data without receiving the intervention. Including a CSC in an experimental design ‘provides additional control and strengthens the between series comparisons’ ([55], p. 267). Of the eight intervention participants, three had a primary diagnosis of ASD, two ADHD, and three had an SLD (see Table 1). The inclusion criteria for participation in the intervention were a formal diagnosis, Level 1 functioning, fully integrated into mainstream school classes, and having sufficient literacy skills to participate in the Minds Online intervention program. Level 1 indicates that adolescents with an NDD require minimum support (e.g., help with organization or planning) for their everyday independent functioning in school or with their relationships compared to others of the same age and background. Individuals categorized as Level 2 or 3 require substantial external support such as speech therapy, or a full-time aide to function independently. Participants were excluded from this study if parent/guardian (and their own) consent was not obtained. The ages of the participants ranged from 11 years 1 month to 12 years 3 months (Mean age = 11 years 8 months, SD = 0.43 months). Of the sample, seven were Australian, one male was born in the USA and one female was born in Europe.

The participants were recruited from one co-educational school in Perth, the capital city of Western Australia, with an ICSEA value of 1050. ICSEA is set at an average of 1000 [SD = 100] and the higher the ICSEA value, the higher the level of educational advantage of students, and vice versa [56]. The participating school has students from a variety of ethnic backgrounds who fall within the designation of average educational advantage based on the ICSEA values.

### 3.2. Minds Online: A Serious Game Intervention

Minds Online [57] is a fully interactive 3-D animated serious game that embeds a therapeutic approach (CBM-I training) into a world of 3-D environments, e.g., classroom, schoolyard, home, school bus, using social media (see Appendix A). The primary aim of Minds Online is to alter the negative interpretive bias (to more benign interpretations) in individuals with a cognitive bias towards threatening interpretations during social information processing, particularly in ambiguous social situations. Consisting of 10 × 25 min episodes, Minds Online presents everyday school situations and scenarios (e.g., preparing for a test, taking part in group discussions, meeting friends at lunchtime, working in groups, being on the school bus, using social media to communicate with friends) that can elicit a negative or benign bias response. Minds Online has an overarching story that connects the 10 episodes, with individual stories existing within each separate episode. All 10 episodes are fully narrated, and the words appear as text on screen synchronously with the narration [57].

Participants begin Minds Online by choosing an avatar that represents themselves. They then complete a three-minute tutorial, which introduces Minds Online and provides an explanation of the purpose of the game and how to play it. Participants also complete several exercises within this tutorial to familiarize themselves with the game and its navigation [57].

Each episode has 20 scenarios containing ambiguities. Participants are required to resolve the ambiguities (presented visually in three short sequences, as text, and narrated) by completing the first missing letter of a word fragment that appears in text on screen as quickly as possible. Episodes 1 (pre-game assessment) and 10 (post-game assessment) present ambiguous events that can be interpreted in either a negative or benign way via the word fragment completion. This provides a pre- versus post-program index of negative interpretive bias. In Episodes 2 to 9 (training) participants learn to resolve ambiguous information, interpreted as a threat, but only in a benign way, thereby training benign interpretive bias. All data (e.g., correctly supplying the first missing letter, speed of inserting the first missing letter correctly/incorrectly, number of attempts made to correctly insert letter) are downloaded at the point of performance (i.e., in real time).

The protocol for how many letters should be missing from the word fragments is if a word is made up of ≤5 letters, then one letter is missing in the word fragment presented to participants. Two letters are missing for words of 6–9 letters and three for words comprising ≥10 letters. The first letter of words should never be missing, and adjacent letters should not be missing.

### 3.3. Study Design

#### 3.3.1. Baseline Phase

Following the initial data collection via standardized measures (described later), participants were assigned to either Groups 1 or 2 based on NDD status and the school timetable requirements, resulting in ASD/ADHD and ADHD/SLD groups. Both groups then commenced the baseline phase at the same time. During baseline, the participants completed electronic daily self-report measures assessing worry about schoolwork, worry about friendships, and feelings of loneliness as identified in [58]. The first author generated a random daily time for which participants received a prompt via their electronic devices to provide the self-report information using pictorial thermometers where feelings were illustrated with a range of emojis. Prior to this study, a range of emojis (relating to worry and loneliness) were shown to adolescents with NDDs who were asked to indicate which ones showed the ‘Most’ and the ‘Least’ feelings, along with varying degrees of these. The most frequently chosen emojis were incorporated into three separate pictorial thermometers for worry about schoolwork, worry about friendships, and feelings of loneliness. Self-reporting started on day 1 of baseline. A total of 40 data points were collected during the baseline phase, which represented an overall compliance by participants of 92%.

#### 3.3.2. Intervention Phase (Minds Online)

Following baseline data collection, all participants (except the CSC) received 10 × 25 min episodes of Minds Online. Group 1 began the intervention prior to Group 2 to establish the staggered baseline phase lengths. Two episodes were delivered each week over a period of five weeks. During this phase, the participants continued to self-report their levels of worry related to schoolwork, friendships, and feelings of loneliness at the random daily times in the school day. A total of 192 data points were collected during the intervention phase, which represented an overall participant compliance of 89%.

#### 3.3.3. Maintenance Phase

One week following the cessation of the intervention (Minds Online), the participants were instructed to resume their random daily electronic self-reporting to determine if any changes in self-reported levels of worry and feelings of loneliness had been maintained. No incentives were offered to participants to complete this phase. A total of 38 data points were collected during the maintenance phase, which represented an overall participant compliance rate of 86%.

### 3.4. Measures

#### 3.4.1. Real-Time Interpretive Bias—Minds Online

The objective of Minds Online is to alter the negative interpretive bias a person may have in information processing towards ambiguous situations and/or information, to more benign interpretations. The time it takes to correctly supply the first missing letter of word fragments, consistent with either a negative interpretive bias or with a benign interpretive bias, is measured. Minds Online targets modifying those negative processing thought biases that may remain outside of an individual’s awareness by training them to complete the first missing letter of word fragments as quickly as possible [57].

#### 3.4.2. Pre- and Post-Minds Online Treatment Standardized Measures

Four standardized measures were administered to all the participants online via Qualtrics in the week prior to the baseline pre-treatment phase and again post-treatment following the cessation of the intervention.

#### 3.4.3. The Perth Adolescent Worry Scale (PAWS)

The PAWS [58] is a validated 12-item two-factor measure of Worry about Academic Success and the Future (6 items) and Worry about Peer Relationships (6 items). Co-designed with adolescents with or without NDDs, parents/guardians, teachers, and school psychologists, the initial validation of the PAWS revealed satisfactory model fit: χ^2^ (df = 53) = 222.03, *p* < 0.001; CFI = 0.93; RMSEA = 0.08; and acceptable reliability for both subscales (α Peer = 0.83; α Academic = 0.88). In this present study, the Cronbach’s alpha pre- and post-intervention, respectively, were sufficiently high to provide confidence in the use of the two subscale scores: Worry about Academic Success and the Future (α = 0.84, α = 0.86) and Worry about Peer Relationships (α = 0.92, α = 0.84).

#### 3.4.4. The Perth A-Loneness Scale (PALS)

The self-report PALS is a 24-item validated [59,60,61] measure with a six-point scale (‘Never’ through to ‘Always’) assessing four correlated factors: friendship related loneliness (i.e., having reliable, trustworthy supportive friends); isolation (i.e., having few friends or believing that there is no-one around offering support); positive attitude to solitude (i.e., positive aspects and benefits of being alone); and negative attitude to solitude (i.e., negative aspects of being alone). In the present study, the estimates of reliability pre- and post-intervention, respectively, were: Friendship related loneliness (α = 0.71, α = 0.74), Feelings of isolation (α = 0.80, α = 0.78), Positive attitude to solitude (α = 0.83, α = 0.80), and Negative attitude to solitude (α = 0.70, α = 0.71).

#### 3.4.5. The Warwick-Edinburgh Mental Well-Being Scale (WEMWBS)

The WEMWBS [62] is a positively worded 14-item self-report measure of positive mental wellbeing. Participants respond according to their feelings over the previous two weeks using a five-point scale (ranging from 1 = ‘None of the time’ to 5 = ‘All of the time’), thereby providing a total score of between 14 and 60. In the present study, the Cronbach alphas were: Pre-intervention (α = 0.72); and post-intervention (α = 0.78).

#### 3.4.6. The Multidimensional Anxiety Scale for Children (MASC)

The MASC is a validated [63,64] self-report instrument, which assesses the major dimensions of anxiety in 8- to 19-year-olds. Four subscales of the MASC were used (pre- and post-intervention internal reliabilities, respectively, shown in parentheses): Social Anxiety (α = 0.88, α = 0.71), Physical Symptoms of Anxiety (α = 0.82, α = 0.84), Harm Avoidance (α = 0.72, α = 0.70), and Separation Anxiety (α = 0.66, α = 0.64). The α’s for Separation Anxiety were less than the recommended α = 0.70, which may be a consequence of this subscale having fewer items than the three better performing subscales see [65].

#### 3.4.7. Self-Reported Data

All participants (including the CSC) provided daily self-report data pertaining to worry about schoolwork and friendships, and feelings of loneliness on separate visual thermometers illustrated with facial emojis representing how they might feel at that time. At a randomly generated time throughout each day during the baseline, intervention and maintenance phases, all participants received an electronic reminder from the first author to access their personal reporting document and to indicate how they felt at that time.

### 3.5. Procedure

Ethics approval was granted by the Human Research Ethics Committee of the University of Western Australia (2019/RA/4/20/6130), and the principal of the participating school. Letters describing the research and consent forms were provided to 12 parents of adolescents in Grade 6, who had a diagnosed NDD. Nine parents agreed to be involved. The first author met with the participants and provided information about the game and how they would be self-reporting each day. At this time, the thermometer scales were also described. None of the participants had played Minds Online prior to this study, but like all adolescents had played digital games each day on their laptops or phones.

Prior to the baseline phase, the pre-intervention standardized measures were administered online via Qualtrics. Participants played two episodes of Minds Online per week in their respective group. Prior to beginning (each episode), the participants were informed that if they had any questions at any time, or required help, they were to raise their hand. The first author was present in each session to ensure that the participants interacted appropriately with the Minds Online game, but at no time was additional support requested by the participants. In addition, the software developers were online during each session to monitor participants’ progression through each episode and to provide technical support should it be needed. At the end of each session, the software developers provided feedback to the researcher regarding the participants’ game play. Once participants had completed the three-minute tutorial in Episode 1 of Minds Online, they began the first episode of Minds Online. The intervention ceased once all 10 episodes had been completed. (In Minds Online, players cannot progress to new episodes outside of school hours and until they have completed the previous episode). When Episode 10 was completed, participants completed the same standardized measures as in the pre-test assessment. Self-reporting also ceased at this time, but one week later, the participants resumed for four days using the same thermometer measures as previously used. The CSC did not complete pre- or post-assessments and did not play the game.

### 3.6. Data Analyses

Different types of data were collected in the present study to assess aspects of the Minds Online intervention, which taken together provide an overall assessment of its effectiveness. The primary objective of Minds Online is to alter adolescents’ negative interpretative biases because these are known to precede worry, adverse mental health, and loneliness. Changes in bias are measured via participants’ response times for word fragment completion to negative or benign statements in Minds Online episodes. Given that changes in interpretive bias should be reflected in outcomes such as worry, mental health, and loneliness, standardized measures were also administered. In addition, daily self-report data were collected from participants. This was included to examine trends over each of the baseline, intervention, and maintenance phases and because mean scores can be unduly influenced when a wide range of scores is generated, especially from a relatively small sample. Furthermore, worry, mental health, and loneliness are all subjective dispositions that are experienced by the individual and as such should be reported on by the individual.

There were three components to the data analysis in this study. First, the changes in interpretive bias as measured by the real-time data downloaded as participants played during game play were calculated. For the present study, a ‘negative interpretive bias index’ was computed, that reflects the relative degree to which participants impose negative, compared to benign, interpretations of ambiguous information. This index was computed using measures of participants’ latencies to accurately identify the first missing letter of the word fragments that follow the ambiguity, here termed ‘completion latency’. These fragmented words are consistent with either the negative or benign interpretations of the preceding ambiguity. It is assumed that a word fragment will be resolved more quickly when its meaning is consistent with, rather than inconsistent with, the meaning that the participant imposed on the preceding ambiguity. Therefore, each participant’s negative interpretation bias index score was obtained from Episode 1 and Episode 10 by subtracting their completion latencies for negative fragments from their completion latency for benign fragments. Higher scores indicate greater relative tendency to impose negative interpretations, compared to benign interpretations, on ambiguous information.

Second, the group pre- and post-standardized measure scores for (i) positive mental wellbeing (WEMWBS), (ii) anxiety symptoms (MASC), (iii) loneliness (PALS), and (iv) worry (PAWS) were examined and compared using paired samples *t* tests using SPSS version 27.0 [66]. Initially, a visual inspection was performed to check the symmetry of the data distribution. This identified potential skew and kurtosis and the negative or positive directions in which they occurred. In addition, 95% confidence intervals were calculated. Confidence intervals provide a range with an upper and lower number describing possible values that the mean could be and are therefore useful for communicating the variation around a point estimate and for providing information to assess the clinical usefulness of an intervention.

Third, the self-report data were analyzed. At randomly generated times each day, the participants were requested to (i) *‘Click on the number from 0–10 that most accurately shows how worried you feel about your schoolwork today’*; (ii) *‘Click on the number from 0–10 that most accurately shows how worried you feel about your friendships today’*; and (iii) *‘Click on the number from 0–10 that most accurately shows how lonely you feel today’*. The raw interrupted time series data for each participant were visually inspected for trends in baseline, intervention, and maintenance effects [67]. This visual inspection also provided an indication of replicated treatment effects. The self-reported interrupted time series data trends for each participant across each of the phases was then analyzed using DMITSA 2.0 [68], a statistical program specifically designed for the analysis of interrupted time series data.

## 4. Results

### 4.1. Treatment Adherence

Over the three phases, the participants’ compliance for the electronic self-reporting ranged from 75% (Participant 4) to 100% (Participants 5, 6, and 7), with an average adherence of 90%. In total, 308 self-reported data points were provided during the three phases across 30 days, including those of the CSC who contributed 24 data points. All participants (except the CSC) completed all 10 episodes of Minds Online (100% adherence).

### 4.2. Treatment Fidelity

To ensure that the Minds Online intervention program was delivered as planned and designed, see [69], the authors contacted the programmers who monitored live game play during the intervention to check on game functionality, the participants’ progression, and if necessary to have any issues clarified.

### 4.3. Analysis One: Index of Interpretive Bias

Seven of the eight participants evidenced changes in their interpretive bias, shifting from a pre-Minds Online negative interpretive bias to a post-program benign interpretive bias. Participants 4, 6, and 8 demonstrated the largest reductions in negative interpretive bias to a more benign interpretive bias, followed by participants 1, 2, 3, and 7 who each evidenced significant reductions in negative interpretive bias to a more benign interpretive bias; participant 5 showed relatively small change in interpretive bias with a tendency to interpret ambiguous information with a more negative interpretive bias following participation in Minds Online.

### 4.4. Analysis Two: Pre- and Post-Group Mean Scores on Standardized Measures

Table 2 shows that there were no significant changes in pre- to post-group mean scores in the desired direction for any of the 11 measures. The reduction in pre- to post-program levels of social anxiety (*p* < 0.08) was the only one approaching significance.

### 4.5. Analyses Three: Individual Participant’s Interrupted Time Series Self-Reported Scores

The observational raw interrupted time series data for each participant was visually inspected for trends across baseline, intervention, and maintenance [67]. Given the complexity of the visual trends for the three self-reported measures across three phases for all participants, these are presented individually in Appendix A. As can be seen in Appendix A, the visual trends for worry about schoolwork (blue line), worry about friendships (red line), and feelings of loneliness (green line) are low and stable in parts, and they also show trends that are variable and relatively unstable. These visual trends did not provide evidence of a replicated treatment effect for reduced worry about schoolwork, worry about friendships, and feelings of loneliness across all eight adolescents in the anticipated direction following the introduction of the intervention. In the case of the CSC, the visual trends remained unstable and variable with little change across phases.

Subtle changes between phases may not be readily discernible visually and so treatment effects were examined for individual participants using DMITSA 2.0, a statistical program specifically designed for the analysis of interrupted time series data. Table 3, Table 4, and Table 5 show that for self-reported worry about schoolwork (Table 3), participants 1 and 2 evidenced significant reductions from baseline to intervention, while participant 7 had a significant reduction from the initial baseline to maintenance. For worry about friendships (see Table 4), there were significant increases from baseline to intervention for participants 4, 6, and 8, while participant 3 reported a significant reduction. Participant 6 evidenced a significant reduction from baseline to maintenance, while participant 5 had a significant increase. Table 5 shows that for feelings of loneliness, participant 2 reported significant reductions from baseline to intervention and from baseline to maintenance. Participants 5 (baseline to maintenance) and 8 (intervention to maintenance) also reported significant reductions. Conversely, participants 1 (baseline to maintenance), 3 (intervention to maintenance), and 6 (baseline to maintenance) reported increases.

## 5. Discussion

Compared to their neurotypical peers, adolescents with ADHD, ASD, and/or SLD are at greater risk of mental health problems [70,71], rejection, isolation, and loneliness [72] especially in mainstream secondary schools where interpersonal interactions are known to be complex, see [73]. Research shows that cognitive biases (that promote negative thinking) are involved in the onset and maintenance of psychopathology [74,75,76] and loneliness [77,78]. Therefore, directly tackling the biased interpretive processes that give rise to negative distorted thinking may be an effective way to reduce mental health problems and loneliness.

Cognitive Behavior Therapy (CBT) has been one of the preferred and most widely used therapeutic approaches by school psychologists for addressing maladaptive thoughts in adolescents [79]. However, as we highlighted in the introduction of this study, the level of introspection and ability to be conscious of one’s own negative thoughts, which the person must be able to evaluate and resolve, can often be challenging for children and adolescents [80], especially for those with NDDs, who may also be socially reticent in face-to-face situations [80]. The limited availability of school psychologists, the increasing prevalence of mental health problems, and a crowded school curriculum means accessibility to CBT via school psychologists is limited, see [81].

Embedding therapeutic elements into serious games that not only engage young people but maintain that engagement, see [47,49], provides an attractive alternative for the delivery of school-based programs to large groups. Earlier attempts to incorporate CBM procedures into games, e.g., refs. [82,83] produced mixed results [48,84,85,86]. In Minds Online [57], the 3-D environments were specifically designed to closely match the settings and scenarios in which young people, with or without NDDs, encounter interpersonal ambiguity and threat [87], a critical element missing from earlier games [88]. Ensuring this and utilizing 3-D animation (rather than 2-D as in previous games) may be why there was excellent program adherence (100%) and why seven of the eight participants evidenced positive changes in their interpretive bias. As such, Minds Online seems to be effective in altering the interpretive bias of adolescents with NDDs. This outcome is very promising because it shows that traditional CBM-I translates into real world settings [88].

Changes in negative interpretive bias should lead to reductions in adverse mental health but changes in interpretive bias here were not reflected in the pre- and post-group mean scores on the standardized measures. There were pre- to post-reductions in the desired direction for social anxiety, harm avoidance, and separation anxiety, but none of these were significant. Social anxiety, which is characterized by excessive and irrational worry [6], approached significance, which is encouraging since up to 38% of 6- to 15-year-olds with ADHD and up to 49% with ASD [89,90] experience social anxiety. A recent meta-analysis [91] also showed that traditional CBM-I had no significant effects on mental health outcomes, which is consistent with Cristea et al. [44].

The present study used a highly engaging CBM-I approach yet there was no effect on mental health outcomes. There are several possible reasons for this lack of translation to mental health outcomes. For example, CBM interventions might be only transient or switch off cognitive biases temporarily and therefore have limited power to impact mental health outcomes [44]. More time may be necessary (e.g., during follow-up) for changes in biases to be shown on standardized measures [44]. In the present study, this latter point may be questioned however, since there were 10 × 25 min sessions, compared to most other studies where far fewer sessions have been administered.

The group mean score for worry about peer relationships appeared to increase (not significantly) following participation in Minds Online. Relatedly, quality of friendships seemed to improve (not significantly), and feelings of isolation also increased at the same time. The social cognition difficulties that characterize adolescents with NDDs may have affected their ability to fully understand social interactions in the game, and others’ emotions, intentions, and beliefs as described by [23]. As such, these adolescents still perceived a discrepancy between the quantity or quality of their actual and desired social relationships [92,93], which contributed to worries about peer relationships and apparent increases in feelings of isolation. The opinions and evaluations of peers are salient in the everyday dynamics of young people, especially for reciprocated friendships within the school context, because these are uniquely related to feelings of loneliness and the development of mental health problems [94]. Therefore, it is important to further explore the lived experiences of adolescents with NDDs in terms of their friendships and feelings of isolation to gain insight into how the scenarios in Minds Online that present social interactions with others can be modified to improve effectiveness. For example, not fitting in and being socially rejected by peers are a key part in the expression of the cognitive biases that can lead to feeling lonely and these may not have been accurately represented in Minds Online. However, it is possible that Minds Online is a specific intervention for reducing interpretive bias, and not transdiagnostic (i.e., across bias, mental health, *and* loneliness) and that a separate tailored CBM-I program is required for friendships and loneliness.

All participants, including the CSC, collected self-reported time series data (daily) across all three phases and this revealed significant changes in levels of worry and loneliness at different times during the intervention. Although the data were self-reported by adolescents and, as such, subject to common method bias, recall bias, social desirability bias, and cognitive abilities, adolescents themselves are best placed to report on the subjective dispositions they experience [95]. Furthermore, research shows that adolescents with NDDs (mainly ADHD) are typically aware of their learning and social problems and do not consistently underestimate the extent of their difficulties [96]. The time series data therefore is added proof that Minds Online was effective in some phases in reducing self-perceived levels of worry and loneliness. However, although highly valid standardized self-report measures were used in the present study, along with the time series self-report measures, future evaluations of Minds Online should also obtain information from teachers, teacher aides, and parents. This information should focus on participants overt social interactions and other school related behaviors (e.g., academic performance, participation in group activities) rather than internal subjective feelings.

## 6. Limitations

Apart from the real-time point of performance (serious game play) measures, all other data were self-reported by the adolescents participating in the program. Multiple informants are said to be best for data collection [97]. However, perceiving the internal subjective dispositions of their children (e.g., worry, feelings of isolation) is hard for parents and teachers, while young people are reticent to report their internal states to their caregivers [95].

Although the real-time data highlighted a replicated treatment effect in the desired direction across seven of the eight adolescents, the pre- and post-standardized measures did not reflect such positive change. It may be that the chosen measures were insufficient for assessing symptoms of the dependent variables in young people with NDDs and therefore resulted in limited sensitivity to treatment changes over the different phases. It is also possible that because levels of anxiety, worry, and loneliness were relatively low at baseline (in both the standardized measures and self-report), there was limited room for subsequent reductions over the course of treatment. Future studies might include additional analyses of data that take baseline levels of symptom severity into account when examining treatment effects.

Another potential limitation of the present study is that the NDD sample included adolescents with ADHD, ASD, and/or SLD and so further evaluation with a larger more diverse and aged population of adolescents with NDDs (e.g., communication disorders, neurodevelopmental motor disorders, and intellectual disabilities) is now required across a greater range of schools. In addition, the absence of information about the diagnostic presentation of the NDDs, for example, the subtype presentation for ADHD, may have impacted participants’ engagement with the measures and game play over the duration of the intervention. Future research involving clinical samples where more detailed information pertaining to comorbidity and presentation are available might be advantageous in evaluating the effectiveness of Minds Online.

The implementation of multiple baseline designs in schools always presents potential limitations. Trying to conduct a study in which adolescents are staggered in their introduction to an intervention in a busy timetabled environment can impact the amount of time available and/or lead to irregular data collection. Although this did not happen to any great extent in this present study, it can be a threat to internal validity and is therefore worth noting.

## 7. Implications for Research

Some of the present findings support the assertion that games and gamification technologies offer the potential for innovative, novel, and engaging intervention therapies for young people [49] and provides an alternative approach to enhancing mental health and wellbeing [98,99]. This has considerable implications for education and schools.

For example, the mode of delivery of Minds Online and the ease with which it can be accessed suggests it could prove beneficial to schools, especially in remote and isolated regions where school psychologists and specialist assistance is limited or not available. Providing an engaging evidence-based program such as Minds Online for adolescents on a ‘telehealth’ basis in remote and rural areas (in Western Australia this can be 1500 kms from a metropolitan area) should be evaluated in future research. This should also examine how parents can support and maintain Minds Online and any subsequent changes in interpretive bias in their children, because greater parental involvement enhances progress in young people who receive intervention [100].

Along with educators, school psychologists play a critical role, especially through conducting psychoeducational assessments, monitoring student progress, and providing direct therapeutic services to young people who face a range of psychological problems [101,102,103]. The role of school psychologists in Minds Online should therefore be considered in future research.

Not all young people in schools have official diagnoses for NDDs because of the long waiting times to access health care professionals. Furthermore, it is known that some adolescents with NDDs use social camouflaging to present a non-NDD persona to fit in and live up to other people’s expectation and to be accepted by others see [104,105]. Future research should include additional assessments to address these potential issues.

## 8. Implications for Practice

The findings of this study lend support to [41,106] who championed schools as ideal places for the implementation and evaluation of interventions focusing on mental health and wellbeing. Minds Online offers schools an adolescent self-directed evidence-based program that requires little teacher input, thereby overcoming many time and resource concerns and constraints. In the present study, one teacher implemented Minds Online over 8–10 weeks and obtained high levels of adherence, which were maintained over time. Several participants were diagnosed with ADHD, which is characterized by a persistent and pervasive pattern of inattention and/or hyperactivity-impulsivity that severely impairs daily functioning. Furthermore, ‘time is the ultimate yet nearly invisible disability afflicting those with ADHD’ ([107], p. 337) such that individuals are unlikely to turn up for treatment or stick to a schedule when in treatment. Individuals with ADHD in this study achieved high levels of adherence which is testament to the ability of Minds Online to engage young people with self-regulation difficulties. This is significant because self-regulation is important for mental health and success in school [108,109,110] and is a strong predictor of future academic achievement [111].

A recent rigorous systematic review argued for the inclusion of interventions rooted in CBT, delivered by clinicians, and targeted at secondary school students [112]. However, young people report confidentiality concerns and feelings of stigmatization with this approach [113]. Implementing CBT in schools can also be problematic. Following a recent evaluation of a brief innovative CBT workshop for schools run by clinicians, the conclusion was that ‘implementing this professionally trained and supervised intervention within the existing and planned infrastructure of school-based services will need considerable thought’ ([114], p. 514). Minds Online has the potential to overcome these difficulties. Specifically, it offers a viable therapeutic approach that requires very little teacher input or supervision, that can be delivered to individuals, or small groups or whole classes during regular lesson time with minimal disruption. By providing adolescents with opportunities to immerse themselves in 3-D animated realistic environments and scenarios removes personalized, face-to-face clinician sessions, fosters positive interpretational tendencies [115,116], and leads to a more flexible thinking approach [117].

In the context of practice, it is important to note that although there were changes in interpretive bias in the present study, these did not translate into improvements in mental health and social relationships (i.e., friendships and feelings of isolation). Indeed, some participants reported increased feelings of isolation. This may be because the cognitive strategies involved in regulating emotional responses that surface and improve during adolescence are not yet consolidated and habitual [118]. Providing feedback to participants during Minds Online training might reinforce the desired interpretation of the game-based scenarios presented so that they become habitual in real life, and this should be considered in future iterations.

## 9. Conclusions

The present study is the first to implement and evaluate an innovative 3-D animated serious game that embeds a therapeutic cognitive based approach (CBM-I) to help adolescents with NDDs resolve negative thought patterns that arise from their daily academic and social interactions. It builds on earlier attempts, such as that of MindLight, which was a 2-D animated short duration program without narration and synchronous text, and data download capability. The results from this present multiple-baseline design provides strong evidence of adolescents’ engagement with Minds Online, along with data showing its efficacy in altering negative interpretive bias in real-world contexts. However, the limited translation of these changes to mental health outcomes suggests the need for further research to validate the results in broader and more diverse populations. Prior to any research, consideration should be given to the game-based scenarios that focus on social interactions with others to address possible replications of the unwanted increases that occurred in the present study in feelings of isolation among some participants.

Notwithstanding the limitations in this current study, the early results are promising for CBM-I training embedded in serious games. The next step will be to determine the future of Minds Online as an intervention by conducting a randomized controlled trial where young people completing Minds Online can be compared to a no training control group.

## Figures and Tables

**Table 1 pediatrrep-17-00028-t001:** Demographic information for each participant (Intervention Group 1: Participants 1–4) and (Intervention Group 2: Participants 5–8).

Participant	Sex	Age (In Years and Months)	Primary Diagnosis (Comorbidity in Parentheses)
1	Male	12.0	ASD (ADHD)
2	Male	12.0	SLD
3	Male	11.1	ASD (ADHD)
4	Male	12.3	ASD (ADHD)
5	Female	12.1	ADHD
6	Female	12.0	SLD
7	Male	11.8	ADHD
8	Female	12.1	SLD
9 (CSC)	Male	11.11	ADHD

**Table 2 pediatrrep-17-00028-t002:** Pre- to post-Minds Online standardized measures group mean scores (SD in parentheses).

	Pre	Post			
	$\bar{x}$ (SD)	$\bar{x}$ (SD)	t	*p*	90% CI
Positive Mental Wellbeing	3.35 (0.51)	3.27 (0.59)	0.481	0.77	−0.17, 0.32
Social Anxiety	1.84 (0.70)	1.69 (0.63)	1.52	0.08	−0.08, 0.39
Physical Anxiety	1.27 (0.52)	1.31 (0.57)	−0.351	0.74	−0.27, 0.20
Harm Avoidance	2.04 (0.44)	2.00 (0.39)	0.281	0.79	−0.31, 0.39
Separation Anxiety	1.28 (0.50)	1.10 (0.54)	1.72	0.12	−0.06, 0.42
Quality Friendships	3.90 (0.77)	3.65 (1.0)	0.958	0.36	−0.34, 0.84
Isolation	1.75 (0.82)	2.05 (1.01)	−1.018	0.33	−0.97, 0.37
Positive Attitude	3.65 (0.96)	3.68 (1.04)	−0.118	0.91	−0.67, 0.61
Negative Attitude	3.06 (0.98)	3.15 (1.04)	−0.301	0.77	−0.71, 0.54
Worry Academic/Future	5.87 (4.08)	5.41 (3.63)	0.602	0.56	−1.24, 2.14
Worry Peer Relationships	3.87 (3.40)	4.18 (3.93)	0.799	0.44	−1.21, 0.58

**Table 3 pediatrrep-17-00028-t003:** Phase comparisons for worry about schoolwork based on self-reported interrupted time series data.

Participants	A	B	*p*	B	C	*p*	A	C	*p*
1	6.50	5.8	0.02 *	1.0	2.25	0.28	6.50	2.25	0.37
2	0.50	0.20	0.01 **	0.20	0.60	0.60	0.50	0.60	0.26
3	0.50	0.25	0.21	0.25	0.0	0.54	0.50	0.0	
4	2.0	1.35	0.54	2.0	-		2.0	-	
5	0.75	0.35	0.68	0.35	0.05	0.29	0.75	0.05	0.53
6	2.25	0.80	0.97	0.80	0.0	0.99	2.25	0.0	0.33
7	2.50	1.25	0.67	1.25	0.0	0.52	2.50	0.0	0.05 *
8	7.0	4.16	0.44	4.16	1.75	0.37	7.0	1.75	0.50

A = Baseline, B = Intervention, C = Maintenance. * *p* < 0.05, ** *p* < 0.0; - too much autocorrelation.

**Table 4 pediatrrep-17-00028-t004:** Phase comparisons for worry about peer relationships based on self-reported interrupted time series data.

Participants	A	B	*p*	B	C	*p*	A	C	*p*
1	1.0	1.78	0.37	1.78	2.50	0.74	1.0	0.50	0.15
2	0.0	0.26	0.52	0.26	0.0	0.89	0.0	0.0	-
3	0.50	0.25	0.77	0.25	0.0	0.05 *	0.50	0.0	
4	0.0	0.06	0.05 *	0.06				-	
5	0.0	0.55	0.16	0.55	0.05	0.43	0.0	0.05	0.05 *
6	0.75	0.85	0.01 **	0.85	0.0	0.44	0.75	0.0	0.006 *
7	0.0	1.40	0.82	1.40	1.25	0.35	0.0	1.25	0.46
8	0.0	0.05	0.02 *	0.05	0.0	0.99	0.0	0.0	0.82

A = Baseline, B = Intervention, C = Maintenance. * *p* < 0.05, ** *p* < 0.01; - too much autocorrelation.

**Table 5 pediatrrep-17-00028-t005:** Phase comparisons for feelings of loneliness based on self-reported interrupted time series data.

Participants	A	B	*p*	B	C	*p*	A	C	*p*
1	0.30	0.0	0.15	0.0	0.0	0.92	0.30	0.0	0.01 **
2	0.50	0.0	0.01 **	0.0	0.0	-	0.50	0.0	0.01 **
3	0.0	1.0	0.99	1.0	1.03	0.05 *	1.0	1.0	0.99
4	0.0	0.12	0.005 **	0.12	-		0.0	-	
5	0.25	0.30	0.13	0.30	0.0	0.76	0.25	0.0	0.05 *
6	0.0	0.35	0.71	0.35	0.35	0.58	0.0	0.35	0.01 **
7	1.25	1.70	0.48	1.70	2.75	0.47	1.25	2.75	-
8	0.75	0.53	0.21	0.53	0.0	0.01 *	0.75	0.0	0.92

A = Baseline, B = Intervention, C = Maintenance. * *p* < 0.05, ** *p* < 0.01; - too much autocorrelation.

## Data Availability

Contact the first author for access to the data. The data are not publicly available due to ethical and legal limitations of confidentiality and privacy of protected health information.

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
