# Peer review of "Addressing Cognitive Bias in Adolescents with Neurodevelopmental Disorders Using 3-D Animated Serious Games"

_pediatrrep, 2025, doi:10.3390/pediatric17020028_

Round 1
Reviewer 1 Report
Comments and Suggestions for Authors
1. Introduction:
- You barely mentioned the gap and necessity of this study in the introduction. Please address these issues more detailed.
2. Methods and Materials
- Support statements on Mind Online game by citing references.
Author Response
Reviewer 1
We are grateful to Reviewer 1 for the positive comments.
One of the other reviewers asked us to make extensive amendments and as a result the few amendments requested by the other three reviewers have been mostly subsumed into the other reviewer’s requests. Nevertheless, we have addressed all reviewers concerns separately as best as we can.
Response 1. We have included a paragraph about the gap and necessity of this study in the introduction (penultimate paragraph) as follows:
To date, there appears to be no serious game-based interventions that incorporate daily experiences to directly tackle the biased interpretive processes that give rise to negative distorted thinking, especially among more vulnerable populations such as adolescents with NDDs. Because adolescents with NDDs are socially reticent in face-to-face situations and may not have the level of introspection and ability to be conscious of their own negative thoughts (which they must be able to evaluate and resolve) an engaging serious game intervention may be a viable approach to help these adolescents learn more adaptive interpretative styles that enhance mental health and wellbeing.
This now runs into another reviewer’s request about adding how Minds Online advances the field which we have addressed:
Minds Online, is a serious game intervention and advances the field by offering such an innovative, novel, and engaging approach for delivering a therapy to adolescents with NDDs. Furthermore, by incorporating daily real-life events in game play, Minds Online provides participants with multiple opportunities to visualize themselves experiencing the scenario in a non-negative interpretational tendency. That is, it provides a more adaptable and functional approach to adolescents in the environment in which they operate and experience life events (e.g., school) and where cognitive and behavioral responses must be intuitive [see 45].
We have also cited references in the methods and materials section for Minds Online.
Reviewer 2 Report
Comments and Suggestions for Authors
The introductory part is sufficiently well documented.
The objectives are not clear. They need to be clarified.
The methodological part is well presented.
The results are mostly reduced to comparisons which are not always relevant. However, given the sample structure this is not necessarily a criticism.
The discussions connect the theoretical and the methodological.
The references are relevant.
The study is well written and structured. I hope it will be published and further cited.
Author Response
Reviewer 2
One of the other reviewers asked us to make extensive amendments and as a result the few amendments requested by the other three reviewers have been mostly subsumed into the other reviewer’s requests. Nevertheless, we have addressed all reviewers concerns separately as best as we can.
We are grateful for the very positive comments from Reviewer 2.
Response 1. We have addressed the comment about objective in conjunction with another reviewer’s suggestion. If we included several specific objectives it would duplicate the material that we have now added under measures (as recommended by another reviewer). In addition, the inclusion of a new paragraph immediately prior to the final paragraph in the introduction subsumes some of the suggestions made here. Therefore, we have amended the first sentence in the final paragraph of the introduction as:
This current study introduces Minds Online, a 3-D animated….
And then in the last part of the final paragraph we state “The main objective of the study is to evaluate Minds Online ………
Reviewer 3 Report
Comments and Suggestions for Authors
On page 2, data on the social difficulties of three groups of disorders are presented: it would be better to divide them in relation to the specific disorder: ADHD, ASD, SLD.
Author Response
Reviewer 3
One of the other reviewers asked us to make extensive amendments and as a result the few amendments requested by the other three reviewers have been mostly subsumed into the other reviewer’s requests. Nevertheless, we have addressed all reviewers concerns separately as best as we can.
Response 1. On page 2, we have amended the sentences as requested to present data on the social difficulties of the three groups of disorders in relation to the specific disorders: ADHD, ASD, SLD separately as follows:
As a result, increased difficulties with social interactions that limit the quantity or quality of friendships they develop is experienced by adolescents with ADHD [24], ASD [14, 25], and comorbid SLD [26]. This leads to increased risk of mental health problems in adolescents with ADHD [12,], ASD [14, 15, 26], and SLD [16]. For many adolescents with NDDs feelings of disconnection, social isolation, and loneliness are subsequently felt [27].
Reviewer 4 Report
Comments and Suggestions for Authors
Previous note
I appreciate the opportunity to review this article, as it represents an excellent learning opportunity, deepening the understanding of the subject, stimulating critical reflection on the methodologies used, and contributing to scientific progress in this field. The article has significant potential for publication, provided that some adjustments are made to strengthen the original text. The topic is highly relevant as it addresses the impact of innovative interventions, such as 3D animated serious games, on promoting mental health and well-being among adolescents with neurodevelopmental disorders (NDDs). This approach is particularly significant since young people with NDDs face unique challenges, such as negative interpretative bias, difficulties in socialization, and increased vulnerability to emotional problems. By exploring the effectiveness of accessible and engaging technological tools, the study offers practical and sustainable alternatives for school contexts, particularly in remote areas or those with limited resources. Furthermore, by integrating an evidence-based cognitive approach (CBM-I), the work contributes to an emerging and promising field, standing out as a relevant effort to address gaps in mental health services for this population.
The potential publication of this article allows Pediatric Reports to significantly reinforce the theoretical foundations of the topic under analysis. However, the study's conclusions must be clearly explained and substantiated to ensure that the manuscript offers genuine value. The target audience for this work includes a diverse group of academic readers with varying levels of expertise, making it crucial for the content to be presented, accessible, and well-structured. Maintaining the high standards of Pediatric Reports in this field is essential. Below are the suggested revisions to enrich the manuscript:
Title
The title is a bit long and could be simplified to improve clarity and impact. It should maintain the main focus of the study without losing its essence but with fewer words and greater objectivity. Just some suggestions.
- Improving Mental Health in Adolescents with NDDs through Serious Games
- Addressing Cognitive Bias in Adolescents with NDDs Using Serious Games
- Serious Games to Tackle Negative Interpretive Bias in Adolescents with NDDs
- 3-D Animated Serious Games for Cognitive Bias Modification in Adolescents with NDDs
Abstract
The title of the article places a lot of emphasis on the idea that the game helped reduce both interpretive biases and mental health issues. However, the abstract reveals that the strength of the results lies primarily in the change in interpretive biases, while improvements in mental health (such as concerns about school, interpersonal relationships, and loneliness) were more limited and only became apparent in more detailed individual analyses. Therefore, it would be beneficial to align the title and the abstract more closely. Perhaps the title could highlight the positive impact on cognitive biases more while toning down the notion of a significant reduction in mental health issues, as the results in this area were not as substantial. This would help avoid setting unrealistic expectations about the study's impact.
1. Introduction
It is suggested to further elaborate on certain aspects related to the specific objectives of the research and the methods used, particularly regarding the sample and the study design. This would make the introduction more detailed and provide a clearer explanation of how the study contributes to the advancement of the field.
2. Materials and Methods
2.1 Participants and Setting
2.2. Minds Online: A Serious Game Intervention
2.3. Study Design
2.3.1 Baseline Phase
2.3.2 Intervention Phase (Minds Online)
2.3.3 Maintenance Phase
Due to the sensitivity of the topic, some aspects can be improved together about certain points in the Methodology. That is:
- In the sample characterization, although it is mentioned that the participants had a formal diagnosis of NDD, exclusion criteria are not specified. It is unclear whether there were specific characteristics (such as comorbidities, severity of the disorder, or other factors) that excluded other adolescents from the study, apart from those who did not consent to participate.
- The sample comes from a single school in Perth, Australia, but there is insufficient information about the socioeconomic and cultural diversity of the participants. Knowing whether the sample includes students from different cultural and economic backgrounds may be important for assessing the generalization of the results.
- It is not mentioned whether the participants had prior experience with digital games or the use of interactive technologies. Familiarity with this type of technology could influence how participants interact with the intervention, which could affect the results.
- Although the participants were divided into two groups based on school schedule needs, it is unclear how this division was made or if there were any specific criteria for the formation of these groups. It would be interesting to understand whether the division was random or based on specific characteristics.
- Regarding the procedures, no details are provided on why the three remaining parents did not agree to participate, which could be relevant for understanding the representativeness of the sample.
- The chapter describes how participants interacted with the Minds Online game, but it is not specified how the participants were monitored during the game to ensure they were interacting properly, nor what type of additional support was offered if participants had difficulties with the game.
- In the maintenance phase, it is not specified whether there was any support or incentive to ensure that participants completed this phase or whether external conditions (e.g., school holidays, changes in routine) could have influenced adherence and the results of this phase.
2.4 Measures
2.4.1 Real Time Interpretive Bias – Minds Online
2.4.2 Pre and Post Minds Online Treatment Standardized Measures
2.4.3 The Perth Adolescent Worry Scale (PAWS)
2.4.4 The Perth A-Loneness Scale (PALS)
2.4.5 The Warwick-Edinburgh Mental Well-being Scale (WEMWBS)
2.4.6 The Multidimensional Anxiety Scale for Children (MASC)
2.4.7 Self-reported Data
2.5 Procedure
2.6 Data Analyses
To improve the organization and avoid mixing information in points 2.4 and 2.6, some changes would be useful. Suggestions:
- In section 2.4. Measures, the focus should be exclusively on describing the instruments used and the data collection methods. Information regarding calculations or analyses, such as the interpretive bias index, should be moved to the "Data Analyses" section. For example, in subsection 2.4.1, it would suffice to explain that the program measures interpretive bias in real time by recording response times for completing word fragments. There is no need to go into detail about the calculation of the index or its formula.
- In point 2.4.7, it would suffice to mention how self-reports were collected, explaining that participants indicated their levels of worry or loneliness on a scale of 0 to 10, without including any detail about how these data were analyzed.
- Point 2.6 Data Analyses, can be expanded to include more details on the calculation of the "negative interpretive bias index". Here, it would be important to clearly explain how this index was derived, describing the statistical methods used to calculate and interpret it. Additionally, it would be helpful to detail the procedures used in t-tests, specifying the statistical criteria applied to compare pre-and post-intervention results.
- To better integrate all this information, it would be interesting to include an introduction in the "Data Analyses" section, explaining how the different types of data collected (response times from Minds Online, self-reports, standardized measures) contributed to the overall analysis. This would help establish clear connections between the collected data, the analysis methods applied, and the results presented.
3. Results
3.1 Treatment Adherence
3.2 Treatment Fidelity
3.3 Analysis One: Index of Interpretive Bias
3.4 Analysis Two: Pre and Post Group Mean Scores on Standardized Measures
3.5 Analyses Three: Individual Participant’s Interrupted Time Series Self-Reported Scores
4. Discussion
When simultaneously analyzing the Results and Discussion sections and considering the "Limitations" highlighted in the text, it is suggested to address some observed weaknesses more thoroughly. Specifically:
- The Discussion acknowledges that changes in interpretive bias did not translate into significant improvements in standardized measures of anxiety, loneliness, or well-being but does not delve into why this happened. Since these changes are theoretically associated with improvements in mental health, it would be useful to explore this contradiction more deeply and reflect on possible reasons for the lack of a more robust impact.
- The results show that, for some participants, concerns about friendships and feelings of isolation increased after the program, which goes against the initial expectations. The Discussion notes that these changes may be linked to the social cognition difficulties of adolescents with NDDs, but it would be helpful to detail how the program could be adjusted to mitigate these adverse effects and promote more positive outcomes.
- The Discussion argues that adolescents are aware of their difficulties, thus making self-reports valid. However, the risk of bias cannot be ignored, especially since responses may have been influenced by social desirability or other limitations. It might be useful to highlight this issue further and suggest ways to complement self-reports with other types of data.
- Although the Discussion emphasizes that Minds Online was effective in changing interpretive bias, the results suggest that this change was not enough to lead to a significant impact on participants' mental health or social relationships. This gap is not explored in detail, which might give the impression that the overall effectiveness of the program was slightly overestimated.
- Note on guidelines for future publications: It is not advisable to introduce authors in the Discussion, as this section should focus on interpreting and analyzing the results based on the theoretical framework and references presented earlier. Introducing new authors in the Discussion may convey the impression that the arguments are not properly supported or aligned with what has been developed throughout the text. Moreover, it may generate inconsistencies in the structure of the work, shifting the focus from the analysis of results to the introduction of concepts or studies that should have been presented in the Literature Review or theoretical framework.
Limitations
6. Implications for Research
Although the implications are generally aligned with the positive results of the program regarding interpretive bias and accessibility, they do not adequately address the contradictory results and limitations of the standardized measures. To be more consistent with the findings, it would be important to integrate reflections on how to resolve the issues identified in the study before expanding the application of Minds Online.
7. Implications for Practice
The impact on standardized measures was limited, with changes in interpretive bias but no significant improvements in mental health and social relationship measures. The generalization of the results is restricted, as they were obtained from a small and specific group of adolescents with NDDs. Additionally, some participants reported increased concerns and isolation, which were not addressed in the practical implications.
8. Conclusions
Considering the entire body of the text, it seems that the "Conclusions" section deserves a more comprehensive and detailed discussion. It is suggested to mention the limitations identified in the study, the need for further research to validate the results in broader and more diverse populations and reflect on aspects that could be improved in the implementation of the program (such as the issues related to increased concerns and isolation observed in some participants). In doing so, the conclusion could offer a more balanced and thorough perspective on the results and future research directions.
References
- The year is missing in reference 88.
Author Response
Reviewer 4
Response 1. Title: We are grateful to the reviewer for suggesting several excellent titles. We have taken the second title with a slight modification. The new briefer title is more succinct and impactful:
Addressing Cognitive Bias in Adolescents with Neurodevelopmental Disorders Using 3-D Animated Serious Games
Response 2. Abstract: The new title now aligns with the abstract in that it “highlights the positive impact on cognitive biases more while toning down the notion of a significant reduction in mental health issues”. In addition, we have included “reducing negative bias” in the abstract preceding the reference to mental health. We have also started the subsequent sentence about the results for mental health with “However, pre- and post-test standardized measures revealed …..”.
Response 3. As suggested, we have included additional text in the penultimate paragraph of the introduction about how the study contributes to the advancement of the field. This now reads:
Minds Online, advances the field by offering such an innovative, novel, and engaging approach for delivering a therapy to adolescents with NDDs. By incorporating daily real-life events in game play Minds Online provides participants with multiple opportunities to visualize themselves experiencing the scenario in a non-negative interpretational tendency. That is, it provides a more adaptable and functional approach to adolescents in the environment in which they operate and experience life events (e.g., school) and where cognitive and behavioral responses must be intuitive [see 45].
2.1 Participants and Setting
Response 4. Rather than specifying that all participants were ….. we have rewritten the paragraph as inclusion criteria. Although not requested, we have also included text about what Level 1 is, and stated that there were no other exclusion criteria apart from not gaining parental/guardian and child consent:
The inclusion criteria for participation in the intervention were a formal diagnosis, Level 1 functioning, fully integrated into mainstream school classes and having sufficient literacy skills to participate in the Minds Online intervention program. Level 1 indicates that adolescents with an NDD require minimum support (e.g., help with organization or planning) for their everyday independent functioning in school or with their relationships compared to others of the same age and background. Individuals categorized Level 2 or 3 require substantial external support such as speech therapy, or a full-time aide to function independently. Participants were excluded from the study if parent/guardian (and their own) consent was not obtained.
Response 5. We have included information in separate places under the heading participants and setting about the cultural and economic backgrounds as follows:
Of the sample, seven were Australian, one male was born in the USA and one female was born in Europe.
The participating school has students from a variety of ethnic backgrounds who fall within the designation of average educational advantage based on ICSEA values
Response 6. We have provided information regarding participants prior experience with digital games, or the use of interactive technologies as follows:
None of the participants had played Minds Online prior to the study, but like all adolescents they had played digital games each day on their laptops or phones.
Response 7. We have provided information about how the participants were divided into two groups as follows:
Following the initial data collection via standardized measures (described later) participants were assigned to either Groups 1 or 2 based on NDD status and the school timetable requirements, resulting in ASD/ADHD and ADHD/SLD groups.
Response 8. Under the procedure we have now added information about how the participants were monitored as they interacted with Minds Online to ensure they were interacting properly and the type of support that was available if they had difficulties:
The first author was present in each session to ensure that the participants interacted appropriately with the Minds Online game, but at no time was additional support requested by the participants. In addition, the software developers were online during each session to monitor participant’s progression through each episode and to provide technical support should it be needed. At the end of each session the software developers provided feedback to the researcher regarding participants game play.
Response 9. We have clarified that in the maintenance phase, no incentives were offered to ensure that participants completed this phase.
No incentives were offered to participants to complete this phase.
2.4 Measures
Response 10. To improve the organization and avoid mixing information in points 2.4 and 2.6, we have, as suggested, moved the information about the calculation of the interpretive bias index and its formula in subsection 2.4.1 to 2.6 Data analyses. Within the new section this now reads:
There were three components to the data analysis in this study. First, the changes in interpretive bias as measured by the real time data downloaded as participants played during game play were calculated. For the present study, a ‘negative interpretive bias index’ was computed, that reflects the relative degree to which participants impose negative, compared to benign, …..
Response 11. Similarly, - we have moved the analysis information from point 2.4.7 (self-report) to 2.6 Data analyses. This now reads in the new section:
Third, the self-report data were analyzed. At randomly generated times each day the participants were requested to (i) ‘Click on the number from 0 – 10 that most accurately shows how worried you feel about your schoolwork today’; (ii) ‘Click on the number from 0 – 10 that most accurately shows how worried you feel about your friendships today’; and (iii) ‘Click on the number from 0 – 10 that most accurately shows how lonely you feel today’. The raw interrupted time series data for each participant were ……
Response 12. As requested for Point 2.6 Data Analyses this has been expanded to include more details on the calculation of the "negative interpretive bias index". We rewritten the text and included additional text:
For the present study, a ‘negative interpretive bias index’ was computed, that reflects the relative degree to which participants impose negative, compared to benign, interpretations of ambiguous information. This index was computed using measures of participants’ latencies to accurately identify the first missing letter of the word fragments that follow the ambiguity, here termed ‘completion latency’. These fragmented words are consistent with either the negative or benign interpretations of the preceding ambiguity. It is assumed that a word fragment will be resolved more quickly when its meaning is consistent with, rather than inconsistent with, the meaning that the participant imposed on the preceding ambiguity. Therefore, each participant’s negative interpretation bias index score was obtained from Episode 1 and Episode 10, by subtracting their completion latencies for negative fragments from their completion latency for benign fragments. Higher scores index greater relative tendency to impose negative interpretations, compared to benign interpretations, on ambiguous information.
Response 13. We have now included additional text outlining the process followed for the pre-and post-intervention results:
Second, the group pre- and post-standardized measure scores for: (i) positive mental wellbeing (WEMWBS), (ii) anxiety symptoms (MASC), (iii) loneliness (PALS), and (iv) worry (PAWS) were examined and compared using paired samples t tests using SPSS version 27.0 [66]. Initially, a visual inspection was performed to check the symmetry of the data distribution. This identified potential skew and kurtosis and the negative or positive directions in which they occurred. In addition, 95% confidence intervals were calculated. Confidence intervals provide a range with an upper and lower number describing possible values that the mean could be and are therefore useful for communicating the variation around a point estimate and for providing information to assess the clinical usefulness of an intervention.
Response 14. As requested, we have included an introduction in the "Data Analyses" section, explaining how the different types of data collected contributed to the overall analysis. This, as you say, establishes clear connections between the collected data, the analysis methods applied, and the results presented (We are are grateful to the reviewer and have used some of his/her words to introduce the text):
Different types of data were collected in the present study to assess aspects of the Minds Online intervention, which taken together provide an overall assessment of its effectiveness. The primary objective of Minds Online is to alter adolescent’s negative interpretative biases because these are known to precede worry, adverse mental health and loneliness. Changes in bias are measured via participants response times for word fragment completion to negative or benign statements in Minds Online episodes. Given changes in interpretive bias should be reflected in outcomes such as worry, mental health, and loneliness, standardized measures were also administered. In addition, daily self-report data were collected from participants. This was included to examine trends over each of the baseline, intervention and maintenance phases and because mean scores can be unduly influenced when a wide range of scores is generated, especially from a relatively small sample. Furthermore, worry, mental health and loneliness are all subjective dispositions that are experienced by the individual and as such should be reported on by the individual.
Response 15. The "Limitations" have been addressed more thoroughly as should be evident in different places in the discussion section.
We have acknowledged that changes in interpretive bias did not translate into significant improvements in standardized measures of anxiety, loneliness, or well-being and reflected on possible reasons for the lack of a more robust impact as follows:
The present study used a highly engaging CBM-I approach yet there was no effect on mental health outcomes. There are several possible reasons for this lack of translation to mental health outcomes. For example, CBM interventions might be only transient or switch off cognitive biases temporarily and therefore have limited power to impact mental health outcomes [44]. More time may be necessary (e.g., during follow-up) for changes in biases to be shown on standardized measures [44]. In the present study this latter point may be questioned however, since there were 10 x 25-minute sessions, compared to most other studies where far fewer sessions have been administered.
Response 16. We have made suggestions about how the program, in the light of the friendships and feelings of isolation results, might be adjusted:
Therefore, it is important to further explore the lived experiences of adolescents with NDDs in terms of their friendships and feelings of isolation to gain insight into how the scenarios in Minds Online that present social interactions with others can be modified to improve effectiveness. For example, not fitting in and being socially rejected by peers are a key part in the expression of the cognitive biases that can lead to feeling lonely and these may not have been accurately represented in Minds Online. However, it is possible that Minds Online is a specific intervention for reducing interpretive bias, and not transdiagnostic (i.e., across bias, mental health and loneliness) and that a separate tailored CBM-I program is required for friendships and loneliness.
Response 17. In the discussion paragraph being referred to, information about complementing self-reports with other types of data has been added. We have been careful not to overlap the information provided under limitations regarding self-report.
However, although highly valid standardized self-report measures were used in the present study, along with the time series self-report measures, future evaluations of Minds Online should also obtain information from teachers, teacher aides, and parents. This information should focus on participants overt social interactions and other school related behaviors (e.g., academic performance, participation in group activities) rather than internal subjective feelings.
Response 18. The impression that the overall effectiveness of the program was slightly overestimated has been addressed:
The results from this present multiple-baseline design provides strong evidence of adolescent’s engagement with Minds Online, along with data showing its efficacy in altering negative interpretive bias in real-world contexts. However, the limited translation of these changes to mental health outcomes suggests the need…
Response 19. Rather than repeat the contradictory results in the limitations we have focused on how we might resolve the issues identified:
Future studies might include additional analyses of data that take baseline levels of symptom severity into account when examining treatment effects.
Future research involving clinical samples where more detailed information pertaining to comorbidity and presentation are available might be advantageous in evaluating the effectiveness of Minds Online.
Response 20. Implications for Practice
Given some participants reported increased concerns and isolation, which were not addressed in the practical implications, we have added the following text:
In the context of practice, it is important to note that although there were changes in interpretive bias in the present study, these did not translate into improvements in mental health and social relationships (i.e., friendships and feelings of isolation). Indeed, some participants reported increased feelings of isolation. This may be because the cognitive strategies involved in regulating emotional responses that surface and improve during adolescence are not yet consolidated and habitual [118]. Providing feedback to participants during Minds Online training might reinforce the desired interpretation of the game-based scenarios presented so that they become habitual in real life, and this should be considered in future iterations.
Conclusions
Response 21. The conclusion now offers a more balanced and perspective on the results and future research directions:
However, the limited translation of these changes to mental health outcomes suggests the need for further research to validate the results in broader and more diverse populations. Prior to any research, consideration should be given to the game-based scenarios that focus on social interactions with others to address possible replications of the unwanted increases that occurred in the present study in feelings of isolation among some participants.
Notwithstanding the limitations in this current study, the early results are promising for CBM-I training embedded in serious games. The next step will be to determine the future of Minds Online as an intervention by conducting a randomized controlled trial where young people completing Minds Online can be compared to a no training control group.
Response 22. We have inserted the year in reference 88.
Response 23. One new reference has been inserted: 118. Riddleston, L.; Bangura, E.; Gibson, O.; Qualter, P.; Lau, J.Y. Developing an interpretation bias modification training task for alleviating loneliness in young people. Beh Res Therapy 2023, 168, 104380. doi.org/10.1016/j.brat.2023.104380
Round 2
Reviewer 4 Report
Comments and Suggestions for Authors
Dear Authors,
I appreciate the authors' commitment and dedication to improving the article *Addressing Cognitive Bias in Adolescents with Neurodevelopmental Disorders Using 3-D Animated Serious Games*, especially for the highly satisfactory way they incorporated the suggestions to enhance the quality of the content further.
I admire how the authors have deepened the relevance of the study, highlighting not only the effectiveness of *Minds Online* in altering negative interpretative bias but also its importance within the broader context of the challenges faced by adolescents with NDDs.
The study’s careful and reflective approach demonstrates a genuine commitment to providing rigorous evidence on the impact of the serious game on cognitive biases while also contributing to the understanding of the relationships between these biases, mental well-being, and social adaptation.
This work not only makes a significant contribution to the academic literature but also has the potential to influence educational and therapeutic practices by providing an innovative model for interventions in school settings.
I reiterate my gratitude and congratulations for the excellent work and the dedication demonstrated throughout the process. I am confident that this article will be a valuable contribution to the field of study and will inspire future research and discussions on this crucial topic.
Kind regards,
Revisor da Pediatric Reports